# SMARTSWITCH: ADVANCING LLM REASONING BY OVERCOMING UNDERTHINKING VIA PROMOTING DEEPER THOUGHT EXPLORATION

## ABSTRACT

The long chain-of-thought (LongCoT) capability is central to the recent break-throughs achieved by large language models in complex reasoning tasks. However, the accompanying issue of "*underthinking*", where models exhibit shallow reasoning by frequently switching thoughts without sufficient exploration, limits both performance and token efficiency. To address this problem, we propose a simple yet effective reasoning strategy: the SmartSwitch inference framework. This framework can be easily integrated into any large language model as a plug-and-play solution, continuously monitoring the model's reasoning process to detect underthinking and guide it toward deeper exploration of promising but overlooked thoughts. Specifically, the perception module identifies points where thoughts switch and evaluates the potential of the preceding thought using an off-the-shelf process reward model (PRM). If a high-potential thought is found to be prematurely abandoned, the intervention module interrupts the ongoing inference, backtracks to the point before the switch, and inserts a "deepening prompt" to encourage further exploration along that promising path. Extensive experiments on challenging mathematical reasoning benchmarks demonstrate that our method significantly enhances the performance of various large language models of different sizes.

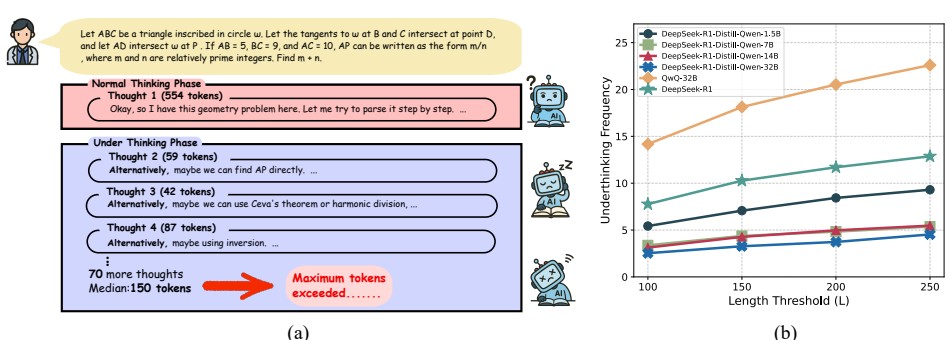

(a)  (b)

Figure 1: Qualitative and Quantitative illustration for the "underthinking problem". (a) presents an example with the underthinking phenomenon sampled from DeepSeek-R1 (Guo et al., 2025). The full response consists of 74 different thoughts, each with a relatively short length (around 150 tokens). (b) shows the "Underthinking Frequency" metric UF($L$) (defined in Eq.(1)) of six mainstream LongCoT LLMs at different values of length threshold $L$. The results show that underthinking is widespread in all models.

## 1  INTRODUCTION

Recent Large Language Models (LLMs) (OpenAI, 2024b; 2025a; DeepMind, 2025; Guo et al., 2025) have demonstrated significant progress, even surpassing human performance on tackling challenging complex reasoning tasks, such as competitive mathematics (AIME, 2024; 2025), programming (Jain et al., 2024), and PhD-level science question answering (Rein et al., 2024). The driving force behind this significant advancement is the Long Chain-of-Thought (LongCoT) reasoning paradigm. Unlike traditional Chain-of-Thought (CoT) reasoning (Wei et al., 2022), LongCoT often

incorporates spontaneous reflection, self-correction mechanisms, and even the ability to switch thinking perspectives (OpenAI, 2024b).

**Observations.** Despite progress, certain issues still limit the performance and efficiency of the LongCoT paradigm, such as the underthinking problem (see Section 3). In particular, models often switch thoughts prematurely without fully exploring their feasibility and potential (see Figure 1). This behavior significantly increases the risk of overlooking promising ideas, ultimately resulting in incorrect final answers. Additionally, frequent thought-switching leads to substantial token wastage.

This underthinking behavior parallels impaired cognitive control in humans, where anxious problem-solvers abandon promising ideas too soon due to low confidence or high perceived failure risk (Robertson et al., 1997; Eysenck et al., 2007). Research shows that external support, like encouraging suggestions or metacognitive prompts from tutors, can help alleviate this tendency (Wells & Matthews, 2016; Clark & Beck, 2011; Cohen et al., 2007; Botvinick & Braver, 2015). These insights emphasize the need for potential assessment mechanisms and confidence calibration to help LLMs avoid underthinking.

**Our Approach.** This paper proposes a novel SmartSwitch inference framework designed to detect and mitigate underthinking in real time. SmartSwitch operates in two cyclical stages. First, the *Perception* module identifies premature thought-switching by detecting linguistic cues (e.g., "Alternatively, ...") that signal a change in direction and evaluates the potential of the just-abandoned reasoning path using a process reward model. Second, if a high-potential thought is deemed to have been prematurely discarded, the *Intervention* module activates. It interrupts the current generation, backtracks to the promising thought, and injects a targeted prompt to encourage deeper exploration along that thought. By enabling the reconsideration of prematurely abandoned yet promising reasoning avenues, SmartSwitch mitigates shallow reasoning and enhances model performance. Furthermore, our framework is fine-tuning-free and plug-and-play, facilitating seamless integration with a wide range of LLMs.

We evaluate our approach on five well-known challenging mathematics benchmarks, including four competition-level datasets — AIME24 (AIME, 2024), AIME25 (AIME, 2025), AMC23 (AMC, 2023), and MATH-500 (Hendrycks et al., 2021), and one standard-level benchmark — GaoKao2023en (Chinese GaoKao Community, 2024). Results in Table 1 show that our SmartSwitch consistently outperforms vanilla inference strategy, and brings significant improvements for existing LLMs with sizes ranging from 1.5B to 32B, demonstrating the good compatibility, generalization, and robustness of our approach. For example, inference by SmartSwitch, the accuracy of DeepSeek-R1-Distill-Qwen-1.5B on AIME24 is boosted by 11.1 points (from 28.9% to 40.0%), and QwQ-32B achieves 73.3% on AIME25 with a gain of 10.0 points.

## 2 RELATED WORK

**Large language models with LongCoT reasoning.** Reasoning ability is a core indicator of the intelligence of Large Language Models (LLMs). For a long time, Chain-of-Thought (CoT) reasoning (Wei et al., 2022) has served as the dominant paradigm, allowing models to reason step by step until deriving the final answer. While effective on many tasks (Cobbe et al., 2021; Chen et al., 2021), CoT-based LLMs still struggle with challenging reasoning problems, for example, GPT-4o (OpenAI, 2024a) achieves only 13.4% accuracy on the well-known AIME24 math competition (AIME, 2024). This landscape changed with the emergence of OpenAI's o1 model (OpenAI, 2024b), which marked a milestone in reasoning LLMs. It demonstrated significant improvements across a wide range of challenging reasoning tasks, including competition-level mathematics (AIME, 2024; 2025), programming (Jain et al., 2024), and PhD-level scientific question answering (Rein et al., 2024). These advances are attributed to a novel reasoning paradigm, Long Chain-of-Thought (LongCoT) reasoning, which enables models to conduct a thorough thinking process before giving a deterministic solution. In contrast to the deterministic reasoning traces in CoT, LongCoT exhibits a more free-form and exploratory structure, allowing the model to explore different ideas, reflect intermediate steps, and correct its own errors. Given its clear advantages, researchers have sought to replicate the capabilities of o1, inspiring a wave of subsequent works, such as closed-source models (DeepMind, 2025), open-source efforts (Guo et al., 2025; Muennighoff et al., 2025; Min et al., 2024; Bespoke Labs, 2025), as well as the upgraded versions from OpenAI itself (OpenAI, 2025a;b).

**Thinking effectiveness in LongCoT reasoning.** Although the LongCoT reasoning paradigm provides opportunities for free and in-depth exploration through a human-like slow thinking phase, the effectiveness of thinking plays a crucial role in determining the performance of the model on challenging reasoning tasks. An effective thinking process can be characterized by several behaviors that involve reasonably planning the reasoning trajectory, for example, reflecting previous steps and exploring new ideas when necessary rather than casually or frequently. The low-effectiveness of thinking in existing LongCoT models (Guo et al., 2025; Qwen Team, 2025; 2024) is reflected in two extremes. On the one hand, the model tends to overthink. Some studies (Chen et al., 2024) have shown that models take about 1000 tokens to reason even for a simple problem like "$1 + 1 = ?$". This redundancy not only leads to unnecessary token usage and inefficient reasoning, but also has no benefit to performance. On the other hand, we found that models still suffer from the underthinking problem. They tend to switch thoughts frequently, e.g., prematurely turning to other thoughts without sufficient exploration on the currect thought. This behavior limits the effectiveness of in-depth thinking and leads to the neglect of promising ideas and the opportunity to derive the correct final answer. Recent study (Wang et al., 2025) also recognized the risk of premature switching and proposed a token-space decoding constraint to suppress the generation probability of tokens corresponding to keywords for switching thoughts. While, such heuristic method introduces artificial bias, which may hinder the indispensable and reasonable exploration behavior due to over-constraining. In contrast, we adaptively steer the model to dive deeper into the current thought or explore a new thought based on the feasibility and potential of the current thought.

## 3 UNDERTHINKING PROBLEM INVESTIGATION

In LongCoT reasoning, a thought refers to an independent reasoning unit aimed at solving a specific sub-problem or achieving an intermediate objective. The model is allowed to switch thoughts when the current thought proves infeasible or the objective itself needs to be redefined. This thought-switching mechanism is a core mechanism, disengaging the model from unproductive explorations and dynamically adapting its reasoning paths.

However, we observe that current LongCoT LLMs often switch thoughts too prematurely before fully exploring the potential of the current thought. This leads to the premature abandonment of promising directions, ultimately harming performance. We refer to this behavior as the "*underthinking problem*". Notably, switching thoughts is not problematic in itself; rather, it is the frequency and hasty switching that undermines deep and effective reasoning.

### 3.1 QUALITATIVE ANALYSIS

Figure 1(a) qualitatively illustrates underthinking in a DeepSeek-R1 response: its reasoning trace exhibits frequent shifts, suggesting insufficient depth. The model prematurely abandons viable strategies (e.g., by partially applying geometric properties like harmonic relations) or disrupts valid reasoning chains through conceptual errors (e.g., conflating distinct geometric points) or misjudgments of problem complexity, resulting in a cascade of short, underdeveloped thoughts.

### 3.2 QUANTITATIVE ANALYSIS

To quantify the underthinking in existing LLMs with LongCoT capabilities, we define a new metric, named Underthinking Frequency, which represents the number of underthinking thoughts in the entire thinking process. Specically, given a LongCoT response consisting of a thinking process $\mathcal{T}$ and a solution $\mathcal{S}$ for a question $Q$, we

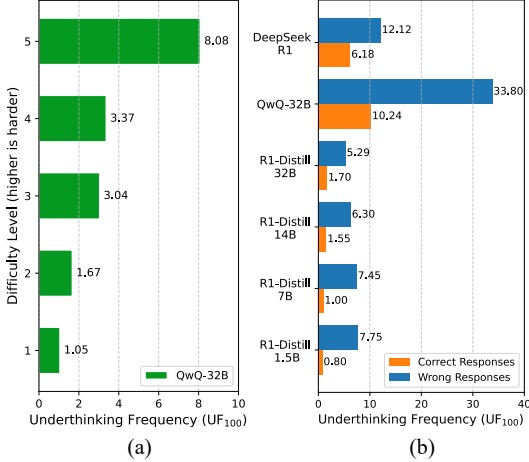

Figure 2: (a) Underthinking frequency increases with problem difficulty on the MATH-500 dataset (Hendrycks et al., 2021). (b) Incorrect answers are associated with a higher frequency of underthinking than correct ones. (Underthinking threshold L=100 tokens; "R1-Distill" is DeepSeek-R1-Distill-Qwen).

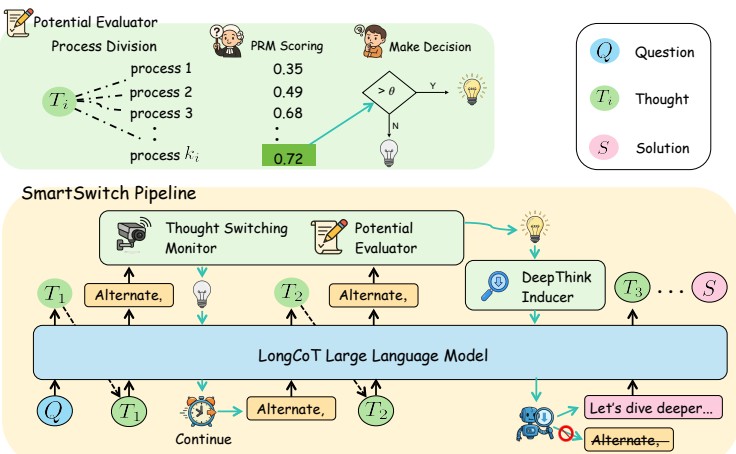

Figure 3: The overall pipeline of the SmartSwitch Inference Framework. During generation, the Perception module monitors for thought switches. When a switch occurs, the preceding thought is evaluated by a Process Reward Model (PRM). If $T_{k-1}$ is deemed promising (score above threshold), the Intervention module activates: generation is paused, the context is reverted to the end of $T_{k-1}$, a "deepen prompt" is inserted, and generation resumes, encouraging deeper exploration of $T_{k-1}$. If not promising, the generation continues.

first segment $\mathcal{T}$ into a sequence of individual thoughts $\{T_i\}_{i=1}^M$, where $T_i$ is the $i$-th thought and $M$ is the total number of thoughts. This segmentation can be performed using a capable LLM (e.g., DeepSeek-V3 (Liu et al., 2024)). The specific prompt used for this process is detailed in Appendix F.3. Then, we can define the *Underthinking Frequency (UF)* metric:

$$\text{UF}_L = \sum_{i=1}^{M} \lambda_i(L), \tag{1}$$

where $\lambda_i(L)$ is a binary variable indicating whether thought $T_i$ exhibits underthinking. Heuristically, we define $\lambda_i(L)$ according to the length of thought $T_i$, that is, $\lambda_i(L) = 1$ if $|T_i| < L$, otherwise $\lambda_i(L) = 0$, where $L$ is the token length threshold.

Figure 1(b) shows the average frequency metric for under-thinking on AIME24 (AIME, 2024) in six main LongCoT LLMs with different values of $L$. Figure 2 illustrates the correlation between underthinking frequency and task difficulty. We conclude three key observations below:

(1) *Prevalence:* All six models consistently exhibit the underthinking behavior, indicating its widespread presence among current LongCoT LLMs.

(2) *Severity:* The degree of underthinking differs across models. QwQ-32B (Qwen Team, 2025) shows the most severe underthinking, while within the DeepSeek-R1-Distill-Qwen series, the smallest 1.5B model exhibits the highest tendency to underthink.

(3) *Contributing Factors:* We observe a clear correlation between underthinking and task difficulty. As Figure 2(a), problems that the model fails to solve tend to trigger more underthinking than those it answers correctly. Underthinking frequency increases steadily with human-annotated difficulty levels, indicating that harder problems tend to amplify underthinking.

## 4 METHODOLOGY

To address the underthinking problem, we propose the SmartSwitch inference framework. This framework aims to dynamically guide LLMs towards deeper exploration of promising reasoning paths that might otherwise be prematurely abandoned.

### 4.1 MOTIVATION

The investigation in Section 3 reveals that LLMs, despite their LongCoT capabilities, often fail to fully explore complex problems due to underthinking—rapidly switching between shallow thoughts.

This behavior limits their ability to solve challenging tasks that require sustained, in-depth reasoning. Human problem-solving often benefits from metacognitive strategies, such as recognizing a promising but underdeveloped idea and consciously deciding to delve deeper. Our framework is inspired by this, aiming to equip LLMs with a similar capability: to perceive when a valuable thought is being neglected and to intervene by prompting a more thorough exploration of that thought. The goal is to transform the default, sometimes erratic, exploration pattern into a more deliberate and productive reasoning process.

## 4.2 SMARTSWITCH INFERENCE FRAMEWORK

The SmartSwitch framework operates iteratively during the LLM's generation process, as illustrated in Figure 3. It consists of two main modules: Perception and Intervention. The complete algorithm is detailed in Appendix D.1.

**Perception module.** During the autoregressive generation process, where the LLM $\mathcal{M}$ produces tokens $t_i \sim P_{\mathcal{M}}(t_i \mid Q, t_{1:i-1})$, the Perception module continuously monitors the output stream.

- *Thought Switch Detection:* It looks for linguistic cues (e.g., "Alternatively") that signal a potential shift away from the current line of reasoning. A comprehensive list of these cues is provided in Appendix D.2.
- *Thought Segmentation:* Upon detecting a switch, the primary unit for evaluation is the entire block of text preceding the cue, which we denote as the thought $T_{prev}$. To ensure that these thoughts remain a manageable length for evaluation, we apply a simple rule: if $T_{prev}$ exceeds a predefined threshold (e.g., 200 tokens), it can be further subdivided at natural breaks like paragraph boundaries (\n\n). Otherwise, the entire $T_{prev}$ is passed to the next stage.
- *Potential Evaluation:* The segmented thought $T_{prev}$ is then evaluated by a pre-trained Process Reward Model (PRM). The PRM outputs a score indicating the quality or potential of $T_{prev}$. If this score exceeds a predefined threshold $\tau_{score}$, it suggests that $T_{prev}$ is a promising reasoning path that has likely been abandoned prematurely.

**Intervention module.** If the Perception module flags $T_{prev}$ as a high-potential, prematurely abandoned thought, the Intervention module activates:

- *Interruption and Backtracking:* The LLM's current generation (which has started on a new thought after the switch) is interrupted. The generation context is rolled back to the state immediately after $T_{prev}$ completes but before the switch occurs.
- *Deepen Prompt Injection:* A predefined "deepen prompt" is appended to the context. An example prompt is: *"Wait, this seems like a promising idea. Let's dive deeper into this reasoning path and not give up easily. Continue exploring this direction thoroughly."*
- *Resumed Generation:* The LLM then resumes generation from this modified context, now guided to further explore $T_{prev}$ instead of switching away. To maintain consistency, the generation proceeds with the original inference parameters.

If the PRM score for $T_{prev}$ is below $\tau_{score}$, no intervention occurs, and the LLM continues with its new thought. This cyclical process of perception and potential intervention continues throughout the generation, aiming to foster deeper exploration when beneficial. A maximum intervention depth or count per problem can be set to prevent excessive looping.

By systematically identifying and reinforcing promising but underdeveloped lines of reasoning, SmartSwitch aims to improve the overall quality and success rate of LLM problem-solving without requiring model retraining.

## 5 EXPERIMENTS

### 5.1 EXPERIMENTAL SETUPS

**Baseline Models.** We apply our SmartSwitch inference framework to a variety of advanced LongCoT LLMs with varying sizes (1.5B to 32B), including DeepSeek-R1-Distill-Qwen-1.5B / 7B / 14B / 32B (Guo et al., 2025) and QwQ-32B (Qwen Team, 2025).

Table 1: Comparison of Vanilla inference and our Perception-and-Intervention (*SmartSwitch*) inference framework on different baseline models. We report the pass@1 Accuracy (averaged on 32 responses) for all benchmarks.

| Models | Inference Framework | Competitional-level | | | | Standard-level |
| | | AIME24 | AIME25 | AMC23 | MATH-500 | GaoKao2023en |
|---|---|---|---|---|---|---|
| DeepSeek-R1-Distill Qwen-1.5B | Vanilla | 28.9 | 20.0 | 67.5 | 83.9 | 72.2 |
| | *SmartSwitch* (ours) | 40.0 (+11.1) | 36.7 (+16.7) | 77.5 (+10.0) | 85.8 (+1.9) | 76.9 (+4.7) |
| DeepSeek-R1-Distill Qwen-7B | Vanilla | 55.5 | 30.0 | 85.0 | 92.8 | 82.6 |
| | *SmartSwitch* (ours) | 66.7 (+11.2) | 53.3 (+23.3) | 92.5 (+7.5) | 93.4 (+0.6) | 84.2 (+1.6) |
| DeepSeek-R1-Distill Qwen-14B | Vanilla | 69.7 | 43.3 | 92.5 | 93.2 | 82.4 |
| | *SmartSwitch* (ours) | 76.7 (+7.0) | 53.3 (+10.0) | 100.0 (+7.5) | 95.2 (+2.0) | 86.0 (+3.6) |
| DeepSeek-R1-Distill Qwen-32B | Vanilla | 72.6 | 46.7 | 90.0 | 94.3 | 85.4 |
| | *SmartSwitch* (ours) | 76.7 (+4.1) | 66.7 (+20.0) | 100.0 (+10.0) | 95.2 (+0.9) | 87.0 (+1.6) |
| QwQ-32B | Vanilla | 79.5 | 63.3 | 97.5 | 95.0 | 85.2 |
| | *SmartSwitch* (ours) | 86.7 (+7.2) | 73.3 (+10.0) | 100.0 (+2.5) | 97.0 (+2.0) | 88.3 (+3.1) |

**Evaluation Benchmarks.** We evaluate the models with our SmartSwitch inference framework on various challenging mathematics benchmarks, since mathematical problem solving is one of the most fundamental tasks for assessing the reasoning ability of LLMs. To ensure comprehensiveness, we consider benchmarks spanning two difficulty levels: competition-level and standard-level. The competition-level set includes AIME24 (AIME, 2024), AIME25 (AIME, 2025), AMC23 (AMC, 2023), and MATH-500 (Hendrycks et al., 2021), which are collected from real human math competitions. The standard-level benchmark, GaoKao2023en (Chinese GaoKao Community, 2024), offers a more routine yet still non-trivial evaluation. We report the pass@1 accuracy averaged on 32 responses for all benchmarks.

**Inference Settings.** For fair comparisons, we apply the same inference settings as each baseline model. In particular, the temperature is set to 0.6, and top-p equals 0.95. The maximum output length is limited to 32768 tokens. We generate 32 responses per query to estimate stable pass@1 accuracy. All the experiments are conducted on NVIDIA A100 GPUs.

**Implementation Details.** In our SmartSwitch inference framework, we employ the off-the-shelf Universal-PRM-7B as our thought scoring model (Tan et al., 2025) to evaluate the promising score of each thought. The reason for this choice is attributed to its capability to assess LongCoT reasoning traces, with support for input lengths up to 32768 tokens, which is a substantial increase over the typical 4096-token

Table 2: Comparison on the "response length (token number)" of models under vanilla inference and our SmartSwitch. We report the average length on AIME24 benchmark. "only correct" corresponds to the problems answered correctly.

| Model | Inference Framework | Response Length (Token Number) | |
| | | All ↓ | only correct ↓ |
|---|---|---|---|
| DeepSeek-R1-Distill Qwen-1.5B | Vanilla | 14973.97 | 6424.33 |
| | *SmartSwitch* | $13486.80_{\downarrow 9.93\%}$ | $6125.78_{\downarrow 4.65\%}$ |
| DeepSeek-R1-Distill Qwen-7B | Vanilla | 14663.03 | 9215.86 |
| | *SmartSwitch* | $14240.07_{\downarrow 2.88\%}$ | $8096.79_{\downarrow 12.14\%}$ |
| DeepSeek-R1-Distill Qwen-14B | Vanilla | 14128.90 | 11195.50 |
| | *SmartSwitch* | $14480.20_{\uparrow 2.49\%}$ | $9433.19_{\downarrow 15.74\%}$ |
| DeepSeek-R1-Distill Qwen-32B | Vanilla | 15375.17 | 12272.28 |
| | *SmartSwitch* | $13188.00_{\downarrow 14.22\%}$ | $10284.33_{\downarrow 16.20\%}$ |
| QwQ-32B | Vanilla | 16924.40 | 14115.48 |
| | *SmartSwitch* | $15939.97_{\downarrow 5.82\%}$ | $13116.87_{\downarrow 7.07\%}$ |

Table 3: Comparison of inference time (min/q) and the time change achieved by *SmartSwitch* on competition-level benchmarks.

| Model | Inference Framework | Avg. Time (min/q) | | |
| | | AIME24 ↓ | AIME25 ↓ | AMC23 ↓ |
|---|---|---|---|---|
| DeepSeek-R1-Distill Qwen-1.5B | Vanilla | 3.23 | 2.69 | 1.10 |
| | *SmartSwitch* | $2.14_{\downarrow 33.7\%}$ | $2.30_{\downarrow 14.5\%}$ | $1.09_{\downarrow 0.9\%}$ |
| DeepSeek-R1-Distill Qwen-7B | Vanilla | 3.31 | 3.35 | 0.90 |
| | *SmartSwitch* | $2.14_{\downarrow 35.3\%}$ | $2.30_{\downarrow 31.3\%}$ | $0.72_{\downarrow 20.0\%}$ |
| DeepSeek-R1-Distill Qwen-14B | Vanilla | 2.57 | 3.22 | 1.29 |
| | *SmartSwitch* | $2.09_{\downarrow 18.7\%}$ | $2.43_{\downarrow 24.5\%}$ | $1.07_{\downarrow 17.1\%}$ |
| DeepSeek-R1-Distill Qwen-32B | Vanilla | 4.87 | 5.27 | 2.12 |
| | *SmartSwitch* | $3.91_{\downarrow 19.7\%}$ | $4.98_{\downarrow 5.5\%}$ | $1.91_{\downarrow 9.9\%}$ |
| QwQ-32B | Vanilla | 5.77 | 6.82 | 3.07 |
| | *SmartSwitch* | $4.97_{\downarrow 13.9\%}$ | $5.67_{\downarrow 16.9\%}$ | $2.77_{\downarrow 9.8\%}$ |

limit of most open-source process reward models. We set the promising score threshold to 0.7, meaning that any thought with a score above this value is considered promising and eligible for deepening intervention. To prevent excessive interventions within a single reasoning process, we cap the number of interruptions at three. Furthermore, as part of our thought segmentation strategy, any thought segment $T_{prev}$ that exceeds a 200-token threshold is first subdivided at natural paragraph breaks before being scored by the PRM.

## 5.2 MAIN RESULTS

**Significant Improvements for Small LLMs.** Our SmartSwitch yields substantial gains for smaller models. As shown in Table 1, DeepSeek-R1-Distill-Qwen-1.5B achieves an accuracy gain of 16.7% on AIME25, and DeepSeek-R1-Distill-Qwen-7B is improved by 23.3% points on AIME25.

**Consistent Gains for Large LLMs.** While larger LLMs have already achieved high performance on challenging benchmarks, SmartSwitch continues to bring consistent and substantial improvements on these strong LLMs. Taking QwQ-32B as an example, our SmartSwitch boosts the accuracy from 79.5% to 86.7% (with 7.2 points gain) on AIME24, and the accuracy from 63.3% to 73.3% (with 10.0 points gain) on AIME25. Remarkably, QwQ-32B even achieves 100% accuracy on AMC23 competition. These results highlight the robustness and broad applicability of our SmartSwitch, even for top-performing models with few improvement room.

**Bridging the Gap Across Model Scales.** Our SmartSwitch can also help narrow the performance gap between smaller and larger model variants. For example, DeepSeek-R1-Distill-Qwen-14B with our SmartSwitch inference surpasses the DeepSeek-R1-Distill-Qwen-32B with vanilla inference on all benchmarks (53.3 *vs.* 46.7 on AIME25). This highlights the potential of our approach for enabling more capable reasoning in resource-constrained scenarios.

## 5.3 FURTHER ANALYSIS

**Efficiency.** Interestingly, our SmartSwitch significantly improves inference efficiency by reducing both total inference time and response length, even while explicitly encouraging deeper thinking. On the AIME24 benchmark, our method shortens the total wall-clock inference time, which comprehensively includes all overhead from PRM scoring and intervention management, by 33.7% for the DeepSeek-R1-Distill-1.5B model and 19.7% for the 32B model (Table 3). Concurrently, the average response length is also reduced by 9.93% and 14.22% for the respective models (Table 2). This dual improvement suggests that our SmartSwitch effectively prunes wasteful reasoning on less fruitful thoughts, thereby focusing computational resources and exploration on more promising directions.

**Mitigate Underthinking.** SmartSwitch significantly reduces the underthinking behavior of LLMs. Specifically, when measuring with a token length threshold of $L = 100$, it not only lowers the Underthinking Frequency metric defined in Equation. (1) (as shown in Figure 4(a)), but also decreases the number of thought switches (as illustrated in Figure 4(b)). This leads to more focused and coherent reasoning trajectories.

**Boost Performance on Failures without Hurting Successes.** Our SmartSwitch improves model performance on challenging problems previously answered incorrectly under vanilla inference, while preserving accuracy on those already solved correctly. For DeepSeek-R1-Distill-Qwen-14B on AIME24, SmartSwitch maintains 100% accuracy on all previously correct answers and successfully recovers 20% of the previously incorrect ones. This demonstrates that SmartSwitch delivers targeted gains without compromising existing capabilities.

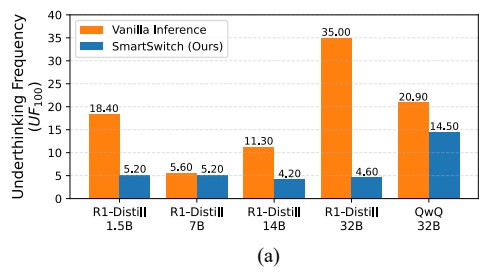

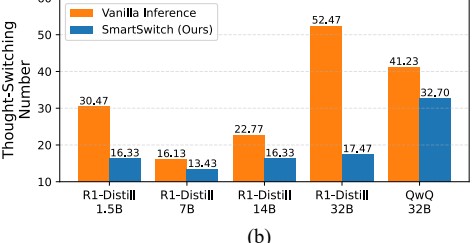

Figure 4: SmartSwitch reduces underthinking frequency and the number of thought-switches on AIME24. "R1-Distill" abbreviates "DeepSeek-R1-Distill-Qwen".

## 5.4 COMPARISON WITH OTHER UNDERTHINKING MITIGATION METHODS

We compare SmartSwitch with two alternative methods for mitigating underthinking:

- *Standard Prompting*: Incorporate general instructions into initial system prompt to encourage deeper thinking "Think step by step. Explore each idea thoroughly before moving on.".

- *TIP (Thought Switching Penalty)* (Wang et al., 2025): A method introduces a penalty on tokens that are associated with thought transitions during decoding.

As shown in Table 5, standard prompting shows nearly no improvement, indicating general instructions are insufficient. TIP only brings limited gain, because it suppresses the decoding probability of the thought-switching tokens indiscriminately, regardless of whether the current thought has become unpromising. This rigid constraint may hinder the model's ability to explore alternative reasoning paths when necessary. In contrast, our SmartSwitch performs best with 40.0% accuracy on AIME24, compared to vanilla inference (28.9%), standard prompting (29.0%), and TIP (31.3%).

## 5.5 ABLATION STUDY

**Potential Scoring Model.** Table 4 presents the performance of various Process Reward Models (PRMs) on AIME25. To quantify the value of PRM guidance, we test an "Always Intervene" baseline that injects a prompt at every thought switch, while adhering to the same three-intervention limit per problem. This naive strategy degrades performance to 18.9%, highlighting the critical role of selective, PRM-guided intervention. Among the PRMs, Universal-PRM-7B achieves the best accuracy at 36.7%. We select it not only for its superior performance but, more importantly, for its essential long-context capability, supporting inputs up to 32,768 tokens. This feature is crucial for evaluating our LongCoT traces and is a key limitation of other PRMs, which either perform worse or lack the necessary context length (see Appendix D.2 for details).

**Process Division Strategy.** To enable effective scoring by the Process Reward Model (PRM), the full reasoning trace must first be divided into coherent processes. Here, we explore four strategies:

- *Model Division (v1)* utilizes a powerful LLM (such as DeepSeek-V3 (Liu et al., 2024)) to perform this division using a carefully designed prompt. This approach introduces additional computational or API cost.

- *Grouped Paragraph (v2)*: This method segments at paragraph boundaries (\n\n) and then groups these initial segments into fixed-size chunks (e.g., five steps).

- *Single Paragraph (v3)*: Segments the output strictly at each detected paragraph boundary (\n\n), treating every resulting block as an individual reasoning step, which can lead to fragmentation.

- *Adaptive Paragraph (v4) (ours)*: Our proposed method (v4) is a multi-stage approach designed to ensure conceptual coherence and optimal segment length for PRM scoring. It first splits the text at logical transition points, such as 'alternate'. If any resulting segment is still longer than 200 tokens, it is further divided using adaptive subdivision–specifically, by breaking at paragraph boundaries (e.g., "\n\n") to maintain readability and structure.

Table 4: Ablation on the effect of different Process Reward Models to scoring the potential.

| Models | Process Reward Model | AIME25 |
|---|---|---|
| DeepSeek-R1-Distill Qwen-1.5B | N/A | 20.0 |
| | Always Intervene | 18.9 |
| | Qwen2.5-Math-PRM-7B | 21.1 |
| | Qwen2.5-Math-7B-PRM800K | 22.3 |
| | Qwen2.5-Math-PRM-72B | 24.8 |
| | Universal-PRM-7B | **36.7** |

Table 5: Comparison of different inference frameworks.

| Model | Inference Framework | AIME24 |
|---|---|---|
| DeepSeek-R1-Distill Qwen-1.5B | Vanilla | 28.9 |
| | Standard Prompting | 29.0 |
| | TIP Wang et al. (2025) | 31.3 |
| | SmartSwitch (ours) | **40.0** |

Table 6: Ablation on the effect of process division strategy on AIME25 benchmark.

| Model | v1 | v2 | v3 | **v4** |
|---|---|---|---|---|
| R1-Distill-Qwen-1.5B | 23.3 | 26.7 | 26.7 | **36.7** |
| R1-Distill-Qwen-7B | 40.0 | 43.3 | 40.0 | **53.3** |
| R1-Distill-Qwen-14B | 43.3 | 46.7 | 46.7 | **53.3** |
| R1-Distill-Qwen-32B | 50.0 | 53.3 | 53.3 | **66.7** |
| QwQ-32B | 70.0 | 70.0 | 73.3 | **73.3** |

As shown in Table 6, strategy v4 consistently outperforms its counterparts (v1, v2, and v3) across all model scales, achieving superior accuracy. The effectiveness of v4 arises from its principled design, which ensures conceptual coherence within each step and optimizes segment length for effective PRM scoring, thereby avoiding the fragmentation issues of strict paragraph splits (v3), the potential conceptual merging of arbitrary grouping (v2), and the additional computational cost and potential inconsistencies of a model-based approach (v1). These results highlight the critical role of a carefully designed step division strategy in maximizing the performance of the framework.

**Process-to-Thought Score Mapping Strategy.** Since the PRM assigns a potential score to each individual process, but a single thought may consist of multiple processes, we need to aggregate these process-level scores to obtain a final score for each thought. We explore several aggregation strategies, including taking the mean, maximum, median, weighted average, or simply the score of the last process within the thought. As shown in Table 7, for a thought, the simple strategy that treating the score of last process within this thought as its final potential score achieves the best performance. Thus, we use this strategy by default.

**Potential Score Threshold.** We investigated the impact of the potential score threshold on R1-Distill-Qwen-1.5B's AIME24 performance (Table 8). Compared to the vanilla baseline (28.90% accuracy), thresholds of 0.68 and 0.69 increased accuracy to 30.00%. Performance peaked significantly at a 0.70 threshold with 40.00% accuracy, before dropping to 30.00% at 0.71. This demonstrates that while a suitable threshold range improves results, selecting the optimal value, such as 0.70 in this case, is crucial.

Table 7: Ablation on the effect of different process-to-thought score mapping strategies.

| Models | Mapping Strategy | AIME24 |
|---|---|---|
| DeepSeek-R1-Distill Qwen-1.5B | max | 33.33 |
| | min | 30.00 |
| | mean | 30.00 |
| | median | 33.33 |
| | weighted average | 33.33 |
| | **last** | **40.00** |

## 6 DISCUSSION

**Limitations.** The efficacy of our framework depends on the quality and calibration of the external Process Reward Model. Its performance is fundamentally bounded by the PRM's ability to accurately assess the potential of diverse reasoning paths. Furthermore, SmartSwitch relies on several key hyperparameters, such as the potential score threshold and the maximum intervention count. While our experiments show that a well-chosen setting is effective across various models, these parameters may require domain-specific or model-specific tuning for optimal performance. Finally, our current thought-switch detection mechanism is based on linguistic cues, which may not capture all instances of premature abandonment, especially those that occur without explicit textual markers. This reliance on explicit markers means it may miss more subtle or implicit shifts in reasoning strategy.

Table 8: AIME24 ablation on the potential score threshold.

| Model | vanilla | 0.68 | 0.69 | **0.70** | 0.71 |
|---|---|---|---|---|---|
| R1-Distill-Qwen-1.5B | 28.9 | 30.0 | 30.0 | **40.0** | 30.0 |
| R1-Distill-Qwen-7B | 55.5 | 53.3 | 43.3 | **66.7** | 43.3 |
| R1-Distill-Qwen-14B | 69.7 | 66.7 | 70.0 | **76.7** | 70.0 |
| R1-Distill-Qwen-32B | 72.6 | 63.3 | 63.3 | **76.7** | 63.3 |
| QwQ-32B | 79.5 | 73.3 | 73.3 | **86.7** | 73.3 |

**Future work.** A primary direction for future work is to reduce the reliance on external components. One promising avenue is to distill the evaluative capabilities of the PRM directly into the base LLM, enabling it to perform self-assessment of its reasoning paths without an external call. This could lead to a more efficient and integrated system. Another area for advancement is the development of more sophisticated intervention mechanisms. Instead of a fixed prompt, a dynamic system could generate context-aware prompts to guide the model's exploration more precisely. Finally, we plan to extend the SmartSwitch framework beyond mathematical reasoning to other complex domains such as software engineering, scientific discovery, and legal analysis, which will require adapting the evaluative criteria and intervention strategies to new contexts.

## 7 CONCLUSION

In this paper, we identify and characterize the "underthinking" phenomenon in LLMs with Long-CoT capabilities, where models prematurely abandon promising reasoning paths, hindering their performance on complex tasks. To address this, we propose the SmartSwitch framework. Using linguistic cues, SmartSwitch detects these switches, employs a PRM to assess abandoned thoughts, and prompts deeper exploration of valuable overlooked paths. This training-free, model-agnostic approach significantly improves LLM performance on mathematical benchmarks by fostering deeper exploration and reducing shallow reasoning. SmartSwitch offers a promising direction for enhancing the reliability and depth of reasoning in LLMs.

## ETHICS STATEMENT

This research adheres to the ICLR Code of Ethics. Our work aims to positively contribute to society by improving the reasoning capabilities of Large Language Models (LLMs), making them more robust and efficient for complex tasks. We acknowledge the importance of the responsible application of this technology. We encourage practitioners who build upon our framework to be mindful of potential societal impacts and to ensure that the underlying models are used in a fair and equitable manner. Our research does not involve the collection or use of new personally identifiable information.

## REPRODUCIBILITY STATEMENT

The supplementary material contains the complete source code to ensure full reproducibility of our results. This encompasses all pipelines used for response generation and the automated evaluation of model outputs.

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

## A    THE USE OF LARGE LANGUAGE MODELS (LLMS)

A large language model was utilized in the preparation of this manuscript to assist with proofreading and improving the clarity of the text. All intellectual content, including ideas, analysis, and conclusions, is solely the work of the authors.

## B    DATASET AND BENCHMARK DETAILS

### B.1    EVALUATION BENCHMARKS

To comprehensively assess the reasoning capabilities of our framework, we conduct evaluations on a curated set of five challenging mathematics benchmarks. These benchmarks span two distinct difficulty tiers: competition-level and standard-level, providing a robust testbed for our method. A detailed description of each benchmark, including its source and the number of questions, is provided in Table 9.

### B.2    MOTIVATION FOR DATASET SELECTION

The selection of these specific benchmarks is motivated by several key criteria. First, they are well-established and widely recognized in the research community for evaluating advanced mathematical reasoning, providing a standardized basis for comparison (Liu et al., 2024; Guo et al., 2025). Second, their public availability is crucial, as it ensures that our experimental results can be independently verified and reproduced by other researchers. All datasets are utilized in strict accordance with their original licenses, and all sources are appropriately cited in the main paper.

Table 9: Details of evaluation benchmarks used in our experiments. All benchmarks are publicly available and selected for their established role in assessing advanced mathematical reasoning.

| Benchmark | Description | # Questions |
|---|---|---|
| *Competition-Level* | | |
| AIME24 (AIME, 2024) | The American Invitational Mathematics Examination 2024, a highly challenging high-school mathematics competition. | 30 |
| AIME25 (AIME, 2025) | The American Invitational Mathematics Examination 2025, continuing the series of challenging problems. | 30 |
| AMC23 (AMC, 2023) | The American Mathematics Competitions, a qualifying competition for the AIME. | 40 |
| MATH-500 (Hendrycks et al., 2021) | A subset of 500 challenging competition-level problems from the comprehensive MATH dataset. | 500 |
| *Standard-Level* | | |
| GaoKao2023en (Chinese GaoKao Community, 2024) | A collection of English-translated mathematics problems from the 2023 Chinese National College Entrance Examination (Gaokao). | 385 |

## C    EXPERIMENTAL SETUP

This section details the experimental configurations used to evaluate our SmartSwitch framework, ensuring full reproducibility.

### C.1    EVALUATION METRIC

Our primary evaluation metric is pass@1 accuracy. To mitigate generation stochasticity, the final score is calculated as the average success rate over 32 independent evaluation runs on the entire benchmark. For correctness, we employ a rigorous automated verifier that checks for mathematical equivalence, thus overcoming the limitations of brittle string matching.

Specifically, we utilize the "symeval" library (Tong, 2024), which implements a robust validation pipeline. This pipeline first extracts the final numerical or symbolic answer from the model's response

using regular expressions and then evaluates its correctness against the ground truth via symbolic comparison with the SymPy library. This method ensures accurate verification for a wide range of mathematical answer formats, including complex numbers, matrices, sets, and symbolic expressions, which would otherwise be prone to evaluation errors. A generation attempt is marked as correct only if the extracted answer is symbolically equivalent to the ground truth.

## C.2 BASELINE MODELS

To demonstrate the generalizability and model-agnostic nature of SmartSwitch, we apply it to a range of state-of-the-art Large Language Models (LLMs) with Long-Chain-of-Thought (LongCoT) capabilities. These models vary in size and architecture, providing a robust testbed for our framework. The specific models are:

- DeepSeek-R1-Distill-Qwen series (1.5B, 7B, 14B, 32B) (Guo et al., 2025)
- QwQ-32B (Qwen Team, 2025)

These models are selected due to their strong baseline performance on reasoning tasks and their publicly available LongCoT generation capabilities.

## C.3 COMPUTING INFRASTRUCTURE AND INFERENCE SETTINGS

**Computing infrastructure.** All experiments are conducted on a cluster of NVIDIA A100 GPUs, each with 80GB of VRAM. The software environment is as follows:

- **Operating System**: Ubuntu 22.04 LTS
- **CUDA Version**: 12.4
- **Python Version**: 3.10
- **Key Libraries**: PyTorch 2.5.1, Transformers 4.53.1, vLLM 0.7.3

**Inference settings.** To ensure fair and reproducible comparisons, we use consistent inference settings across all experiments for both vanilla generation and our SmartSwitch framework.

- **Temperature**: 0.6
- **Top-p**: 0.95
- **Maximum Output Tokens**: 32,768
- **Repetitions per Query**: 32
- **Randomness**: For each of the 32 generations per query, we use a fixed random seed. This ensures that the results are fully reproducible.

---

**Algorithm 1** Simplified PI Framework

**Input:** Original Problem $P_{orig}$; LLM $L$; PRM $R$; Signals $\mathbb{S}$; Deepening Prompt $D_P$; Threshold $\theta$.

**Output:** Final Output $O_{final}$.

1: $L_{input} \leftarrow P_{orig}$
2: **while** True **do**
3:    $O_{current\_pass} \leftarrow$ ""
4:    **for each** $Chunk \leftarrow L.\text{Stream}(L_{input})$ **do**     // LLM streams content
5:       $O_{current\_pass} \leftarrow O_{current\_pass} + Chunk$
6:       **Step 1: Thought Switching Monitor**
7:       $found, O_{prefix} \leftarrow \text{CheckSignal}(\mathbb{S}, O_{current\_pass})$
8:       **if** $found$ **then**
9:          **Step 2: Potential Evaluator**
10:          $score \leftarrow R.\text{Score}(\text{Steps}(O_{prefix}))$
11:          **if** $score \geq \theta$ **then**     // High score: Intervene
12:             **Step 3: DeepThink Inducer**
13:             $L_{input} \leftarrow P_{orig} + O_{prefix} + D_P$
14:             **break**     // Restart pass
15:    **if** finished **then**     // Stream finished naturally
16:       $O_{final} \leftarrow O_{current\_pass}$
17:       **break**     // Process complete
18: **return** $O_{final}$

---

Figure 5: Pseudocode of the SmartSwitch inference algorithm. The framework continuously monitors the generated token stream for thought-switch cues. Upon detection, the preceding thought is evaluated by a Process Reward Model (PRM). If its score exceeds a threshold ($\tau_{\text{score}}$), generation is interrupted and redirected to further explore the promising thought via a "deepen prompt." Otherwise, the model proceeds with the new thought without intervention.

## D SMARTSWITCH FRAMEWORK IMPLEMENTATION

This section provides a detailed description of the SmartSwitch framework's algorithm and its core components.

### D.1 ALGORITHMIC DETAILS

The SmartSwitch framework operates as an intervention loop during the autoregressive generation process. Figure 5 presents the pseudocode for our method. The framework monitors the generated token stream for thought-switch cues. Upon detection, it evaluates the preceding thought using a Process Reward Model (PRM). If the thought is deemed promising (score above $\tau_{\text{score}}$), the generation

is halted, backtracked, and guided to explore the promising thought further by injecting a "deepen prompt." Otherwise, generation continues along the new path.

## D.2 CORE COMPONENT DETAILS

**Thought switch detection.** The framework identifies thought switches by scanning the generated text for specific linguistic cues that indicate the model is abandoning one line of reasoning to start another. Our implementation relies on a predefined set of phrases that signal a deliberate shift in strategy. The complete set of these linguistic cues is detailed in Table 10.

**Process division strategy.** As described in the main paper's ablation study (Section 5.4), we adopt the *Adaptive Paragraph (v4)* strategy for segmenting reasoning traces into processes for PRM evaluation. This method first splits the text at logical transition points (i.e., where a thought switch is detected). If a resulting thought segment exceeds 200 tokens, it is further subdivided at natural paragraph breaks (\n\n). This adaptive strategy ensures that the segments provided to the PRM are both conceptually coherent and within an optimal length for accurate evaluation.

Table 10: The complete set of predefined linguistic cues used to detect thought switches during the reasoning process. The detection of any of these phrases triggers the potential evaluation step.

| Category | Linguistic Cues |
|---|---|
| Simple Alternatives | Alternately, Alternatively, Alternative: Alternative approach: Wait, alternatively, |
| Method/Approach Shifts | Let me try another method Let me try another approach Wait, another approach: Wait, alternate approach: Wait, let me try another method Wait, let me try another approach |

**Potential evaluation with PRM.** The selection of an appropriate Process Reward Model (PRM) is critical for the efficacy of the SmartSwitch framework. The PRM must accurately assess the quality and potential of a given reasoning step to ensure that interventions are both meaningful and beneficial. We considered several state-of-the-art PRMs, each with distinct characteristics:

- **Qwen2.5-Math-PRM (7B & 72B)** (Zhang et al., 2025): This series of models from the Qwen team represents a specialized family of evaluators engineered for process-level supervision in mathematics. In contrast to reward models that only score the final outcome, these PRMs are trained to assess the correctness of intermediate steps within a complex reasoning chain, and they have demonstrated high efficacy on standard error-identification benchmarks.

- **Qwen2.5-Math-7B-PRM800K** (Zhang et al., 2025; Zheng et al., 2024): This model provides a publicly accessible baseline, created by fine-tuning the Qwen2.5-Math-7B-Instruct model on the open-source PRM800K dataset. Its transparent and reproducible training on a well-known corpus makes it a valuable reference point for comparative analysis, though its architecture and training data are less specialized than those of the flagship Qwen PRMs.

- **Universal-PRM-7B** (Tan et al., 2025): This state-of-the-art PRM, also built upon the Qwen2.5-Math-7B-Instruct foundation, was developed externally with a distinct and sophisticated training methodology. Its training regimen integrates techniques such as diverse policy sampling and reverse verification, which were specifically designed to enhance its robustness and generalization. This allows it to accurately score a wider spectrum of reasoning strategies, even those not seen during its training.

Our decision to employ Universal-PRM-7B as the primary thought evaluator is based on three decisive advantages. First, it demonstrates state-of-the-art performance on public leaderboards, achieving a top-tier average score of 74.3 on ProcessBench (Zheng et al., 2024), outperforming other candidates. Second, and critically for our application, it supports a long-context window of up to 32,768 tokens, a necessity for evaluating the extensive reasoning traces generated in LongCoT paradigms. This capability is absent in many other PRMs, which are often limited to 4096 tokens. Finally, its superior benchmark performance was empirically validated in our own ablation studies (see Table 6 in the main paper), where it consistently yielded the highest final task accuracy when integrated into the SmartSwitch framework.

# E    ADDITIONAL EXPERIMENTAL RESULTS AND ANALYSIS

This section provides supplementary results that further substantiate the claims made in the main paper.

## E.1    IMPACT ON CORRECT VERSUS INCORRECT ANSWERS

An analysis of the framework's impact on individual problem outcomes reveals that its performance gains are achieved without compromising existing model capabilities. SmartSwitch primarily improves accuracy by enabling the model to "recover" solutions for problems that it previously answered incorrectly. For instance, when applied to the DeepSeek-R1-Distill-Qwen-14B model on the AIME24 benchmark, our framework successfully converted 20% of the previously incorrect attempts into correct solutions.

Crucially, this improvement does not come at the expense of existing strengths. The framework preserved a 100% success rate on the subset of problems that the baseline model already answered correctly. This demonstrates that SmartSwitch functions as a targeted and non-destructive enhancement, selectively improving performance on challenging problems without introducing negative side-effects on established capabilities.

# F    PROMPT DETAILS

This section provides the exact prompts used in our framework and for baseline comparisons, ensuring full transparency and reproducibility.

## F.1    DEEPEN PROMPT FOR SMARTSWITCH

This prompt is injected by the Intervention module to encourage deeper exploration of a promising thought.

> **Prompt for dividing steps**
>
> ```
> Wait, this seems like a promising idea. Let's dive deeper into this reasoning path and
> not give up easily. Continue exploring this direction thoroughly.
> ```

## F.2    PROCESS REWARD MODEL PROMPTS

**Universal-PRM-7B prompt.**    This is the template used to score a reasoning process with Universal-PRM-7B (Tan et al., 2025).

> **Prompt for Universal-PRM-7B**
>
> ```
> ## System message
> You are a helpful assistant.
>
> ## User query
> {{question}}
> The reference answer is: There is no reference answer for this question.
>
> ## Assistant response:
> <Special-Token> <thought_1> <Special-Token>
> <Special-Token> <thought_2> <Special-Token>
> ...
> <Special-Token> <thought_n> <Special-Token>
> ```

**Qwen-PRM prompt (for ablation).**    This template was used with Qwen-PRM models in our ablation studies. Due to its shorter context limit, a pairwise comparison strategy was adopted.

---

**Prompt for Qwen PRM**

```
## System message
Please reason step by step, and put your final answer within \boxed{}.

## User query
{{question}}

## Assistant response:
<Special-Token> <thought_1> <Special-Token>
<Special-Token> <thought_2> <Special-Token>
...
<Special-Token> <thought_n> <Special-Token>
```

### F.3   PROMPT FOR PROCESS DIVISION

This prompt was used in our ablation study for the *Model Division (v1)* strategy, where a powerful LLM like DeepSeek-V3 is asked to segment the reasoning trace.

---

**Prompt for dividing steps**

```
You are an expert in analyzing and decomposing complex problem-solving processes,
especially in mathematics.

---

Task:

Your task is to divide a long and systematic thinking process (provided below) into
coherent, sequential steps. Each step should represent a complete phase of reasoning,
such as problem analysis, exploration, reassessment, or verification. Ensure **no
content is omitted** between steps, and the entire process is covered from start to
finish.

---

Output Format:
Present the steps in the following structured XML-like format:

```XML
<step number="step id">
    <objective> Purpose of this step </objective>
    <start> First exact sentence of this step in the given thinking process </start>
    <end> Last exact sentence of this step in the given thinking process </end>
</step>
```

---

Key Requirements:
1. **Continuity Preservation**:
   - The `end` sentence of step *i* must **immediately precede** the `start` sentence
   of step *i+1* in the original text.
   - No sentences should be skipped or omitted between steps.

2. **Complete Coverage**:
   - The last step's `end` must be the **very last sentence** of the entire thinking
   process.

3. **Step Objectives**:
   - Label each step's purpose clearly (e.g., "Initial analysis," "Error correction,"
   "Explore different ideas").
   - For backtracking/reassessment, use objectives like "Re-evaluating approach due to
   X."

---

Strict Validation Rules:
1. **Text Continuity Check**:
   - For all steps except the last, the `end` of step *i* must be the **direct
   predecessor** of the `start` of step *i+1* in the original text.
   - Example: If step 1 ends with *"Now I'll try Method A,"* step 2 must start with the
   **very next sentence** in the original text (e.g., *"First, I apply Method A to the
   equation..."*).
```

---

```
2. **Final Step Coverage**:
   - The `end` of the final step **must match** the last sentence of the entire
   thinking process.

---

Instructions:
1. **Read the entire thinking process carefully**: Identify logical segments where the
problem-solver shifts focus (e.g., from analyzing to solving or reflecting, or
exploring, or summarizing).
2. **Define each step**: Assign a unique step number and describe its purpose
(objective).
3. **Adjust step granularity adaptively**: Smaller steps for detailed reasoning, larger
steps for broader phases.
4. **Extract the text**: Mark the exact beginning and ending sentences of each step in
the original text.
5. **Ensure every sentence is included** in exactly one step, with no overlaps or gaps.
6. **Explicitly verify** the key requirements above before finalizing the output.

---

Thinking Process to Decompose (Input):
{{thinking_process}}
```

## F.4 PROMPT FOR TIP BASELINE

This is the prompt template used to generate responses for the Thought Switching Penalty (TIP) baseline (Wang et al., 2025), which includes an instruction to encourage persistence.

> **Prompt engineering for fostering "thought persistence"**
>
> ```
> <context>
> You are an expert math-solving assistant who prioritizes clear, concise solutions. You
> solve
> problems in a single thought process, ensuring accuracy and efficiency. You seek
> clarification
> when needed and respect user preferences even if they are unconventional.
> </context>
>
> <solving rules>
> - Try to complete every idea you think of and don't give up halfway
> - Don't skip steps
> - Display solution process clearly
> - Ask for clarification on ambiguity
> </solving rules>
>
> <format rules>
> - Use equations and explanations for clarity
> - Keep responses brief but complete
> - Provide step-by-step reasoning if needed
> </format rules>
>
> PROBLEM: {{problem}}
>
> OUTPUT: Following above rules to get the correct answer for PROBLEM. Focus on clear,
> concise
> solutions while maintaining a helpful, accurate style.
> ```

## G QUALITATIVE CASE STUDIES

We present three case studies to provide qualitative insight into the operational dynamics of the QwQ-32B model with vanilla inference versus our SmartSwitch framework. These examples illustrate how SmartSwitch mitigates underthinking to improve solution accuracy and efficiency.

In the first case, an AIME25 geometry problem, the vanilla model exhibits clear underthinking (Figure 6). It generates 31,812 tokens and cycles through 126 distinct thoughts but fails to explore promising ideas like the nine-point circle properties, leading to an incorrect answer. In contrast, the SmartSwitch-augmented model solves the problem correctly using only 22,580 tokens. Our

framework intervenes when a promising thought about the nine-point circle (potential score: 0.711) is about to be abandoned, prompting deeper exploration and guiding the model to the correct solution.

The second case, a MATH-500 problem involving parenthesization (Figure 7), further highlights the benefits. The vanilla model consumes 25,469 tokens and undergoes 198 thought switches, yielding an incorrect count. With SmartSwitch, the model correctly identifies all distinct values using 20,488 tokens. The framework intervenes multiple times (e.g., with PRM scores of 0.705, 0.707) to prevent the model from abandoning a systematic exploration, leading to a more robust and efficient reasoning process.

Our final case study on a MATH-500 recurrence relation (Figure 8) shows SmartSwitch's ability to improve efficiency even when the base model is correct. The vanilla model finds the right answer but requires 11,244 tokens and 22 thought switches. The SmartSwitch-augmented model also arrives at the correct answer but does so using only 6,012 tokens. Interventions help consolidate the reasoning path, significantly reducing redundant exploration and demonstrating the framework's value in optimizing the reasoning process.

[AIME25]

Suppose triangle ABC has angles angle BAC = 84 degrees, angle ABC = 60 degrees, and angle ACB = 36 degrees. Let D, E, and F be the midpoints of sides BC, AC, and AB, respectively. The circumcircle of triangle DEF intersects BD, AE, and AF at points G, H, and J, respectively. The points G, D, E, H, J, and F divide the circumcircle of triangle DEF into six minor arcs. Find arc DE + 2 * arc HJ + 3 * arc FG, where the arcs are measured in degrees.

**Vanilla: (31812 tokens)**

**Thought1**: Okay, so I need to find the measure of the sum overarc DE plus twice overarc HJ plus three times overarc FG. ...

First, maybe I should draw a rough sketch or at least visualize the problem ... **So maybe properties of the nine-point circle can be useful here.**

**Thought2**: *Alternatively,* maybe coordinate geometry could work here? Let me see. ...
**[omitted 483 tokens]**

**Thought3**: *Alternatively,* maybe using coordinates would be a good approach here. Let me try ...
**[omitted 278 tokens]**

**Thought126**: *Alternatively,* ithe arcs DE, HJ, FG might each be 60°, leading to 60+120+180=360. ...
**[omitted 44 tokens]**

    •
    •
**Solution:**
The arcs between the points G, D, E, H, J, and F were measured. The arcs DE, HJ, and FG were found to be:

        72 , 130.7 , and 72

The problem asks for the value of:

        arc DE + 2 × arc HJ + 3 × arc FG

Substituting the values:

        72 + 2 × 130.7 + 3 × 72 ≈ 549.4

However, by analyzing the geometric configuration and using properties of the nine-point circle, it can be shown that the exact answer is:

        180 degrees (Wrong !!!)

Final answer: 180 degrees

**SmartSwitch(Our): (22580 tokens)**

**Thought1**: Okay, so I need to find the measure of the sum overarc DE plus twice overarc HJ plus three times overarc FG. ...

First, maybe I should draw a rough sketch or at least visualize the problem ... **So maybe properties of the nine-point circle can be useful here.** *Alternatively,*
**[potential score : 0.711 ]** *Dive Deep !!!*

**Thought2**: **Wait, let's dive deeper into this idea.** The nine-point circle has a radius half of the circumradius of ABC ....

so maybe coordinate system can be chosen so that point B....

Wait, but according to the central angles between the points, **we have to consider the positions of G, H, J.** *Alternatively,*
**[potential score : 0.708 ]** *Dive Deep !!!*

**Thought3**: **Wait, let's dive deeper into this idea.** The arcs between the six points (G, D, E, H, J, F) ....

    •
    •
**Solution:**

DE : 2 × 36° = 72°
EF : 2 × 84° = 168°
FD : 2 × 60° = 120°

Intersection Points and Arcs :

Arc HJ ≈ 24°
Arc FG ≈ 72°

 Final Calculation :

expression
        arc DE + 2 × arc HJ + 3 × arc FG
Substituting the values:
        72° + 2 × 24° + 3 × 72°  = 336°

Final answer: 336 degrees

Figure 6: Case study 1 (AIME25 geometry). Left: vanilla QwQ-32B (incorrect, 31,812 tokens). Right: SmartSwitch-augmented QwQ-32B (correct, 22,580 tokens).

**[Math 500]**

The expression 2 * 3 * 4 * 5 + 1 is equal to 121, since multiplication is carried out before addition. However, we can obtain values other than 121 for this expression if we are allowed to change it by inserting parentheses. For example, we can obtain 144 by writing (2* ( 3 * 4 ) ) * ( 5 + 1 ) = 144.

In total, how many values can be obtained from the expression 2 * 3 * 4 * 5 + 1 by inserting parentheses? (Note that rearranging terms is not allowed, only inserting parentheses.)

**Vanilla: (25469 tokens)**

**Thought1**: *Okay, so I need to figure out how many different values I can get from the expression 2·3·4·5+1 by just inserting parentheses. The original expression is 2\*3\*4\*5+1, which is calculated as ((2\*3)\*4)\*5 + 1 = 120 +1 = 121. ...*

So the operations are:\n\nFirst, 2 * 3,\n\nthen that result *4,\n\nthen that result *5,\n\nthen add 1. ...

Since multiplication has higher precedence than addition, it is evaluated first, so:\n\n( ( (2 * 3) * 4 ) * 5 ) + 1 = 120 +1 = 121. ... **So that changes the order**.

**Thought2**: **Alternatively,** maybe we can group the addition with an earlier multiplication. Let's see.
**[omitted 303 tokens]**

**Thought3**: **Alternatively,** maybe I can model this as an expression tree ...
**[omitted 55 tokens]**

•

**Thought198**: **Alternatively,** if we do (2*(3 +4*5 +1)), but that requires two additions.
**[omitted 23 tokens]**

•
•

**Solution:**

1. Original Expression:
     2 * 3 * 4 * 5 + 1 = 121

2. Grouping the addition with 5 and 1 first:
     2 * 3 * 4 * (5 + 1) = 2 * 3 * 4 * 6 = 144

3. Grouping the addition with 4, 5, and 1:
     2 * 3 * (4 * 5 + 1) = 2 * 3 * 21 = 126

4. Grouping the addition with 3, 4, 5, and 1:
     2 * (3 * 4 * 5 + 1) = 2 * 61 = 122

5. Grouping the addition with 2, 3, 4, and 1, then multiplying by 5:
     (2 * 3 * 4 + 1) * 5 = 25 * 5 = 125 (Wrong !!!)

Final Answer: 5 distinct values in total.

**SmartSwitch(Our): (20488 tokens)**

**Thought1**: *Okay, so I need to figure out how many different values I can get from the expression 2·3·4·5+1 by just inserting parentheses. The original expression is 2\*3\*4\*5+1, which is calculated as ((2\*3)\*4)\*5 + 1 = 120 +1 = 121. ...*

So the operations are:\n\nFirst, 2 * 3,\n\nthen that result *4,\n\nthen that result *5,\n\nthen add 1. ...

Since multiplication has higher precedence than addition, it is evaluated first, so:\n\n( ( (2 * 3) * 4 ) * 5 ) + 1 = 120 +1 = 121. ... **So that changes the order**. ~~Alternatively,~~
**[potential score : 0.705 ]** *Dive Deep !!!*

**Thought2**: **Wait, let's dive deeper into this idea.** The problem allows inserting parentheses ....

So the key is that the operations (multiplications and addition) are in the order they are written, but **we can parenthesize them in different ways.** ~~Alternatively,~~
**[potential score : 0.707 ]** *Dive Deep !!!*

**Thought3**: **Wait, let's dive deeper into this idea.** The expression can be thought of as a list of operations between the numbers and the +1: .... ~~Alternatively,~~
**[potential score : 0.712 ]** *Dive Deep !!!*

•

**Solution:**

1. Original Parenthesization:
     (((2 * 3) * 4) * 5) + 1 = 121

2. Grouping the Addition with 5 and 1:
     (2 * 3 * 4) * (5 + 1) = 2 * 3 * 4 * 6 = 144

3. Grouping the Addition with 4, 5, and 1:
     2 * 3 * (4 * 5 + 1) = 2 * 3 * 21 = 126

4. Grouping the Addition with 3, 4, 5, and 1:
     2 * (3 * 4 * 5 + 1) = 2 * 61 = 122

Thus, the number of distinct values is 4.

Figure 7: Case study 2 (Math 500 parentheses). Left: vanilla QwQ-32B (incorrect, 25,469 tokens). Right: SmartSwitch-augmented QwQ-32B (correct, 20,488 tokens).

**[Math 500]**

If f(n + 1) = (-1)^(n + 1) * n - 2 * f(n) for n >= 1, and f(1) = f(1986), compute f(1) + f(2) + f(3) + ... + f(1985).

**Vanilla: (11244 tokens)**

**Thought1**: *Okay, so I need to solve this recurrence relation problem....*

*Starting with n = 1:\nf(2) = (-1)^{1+1} * 1 - 2f(1) = (-1)^2 * 1 - 2f(1). ... Hmm, this is getting a bit messy, but maybe there's a pattern here. ... Now let's look at the constant terms:\n\nFor n=1: ...Therefore, equate left and right:\n\nC(n+1) + (-2)^n f(1)...* **then maybe relate it using the recurrence?**

**Thought2**: **Alternatively,** *maybe I can compute the sum S = sum_{k=1}^{1985} f(k) in terms of f(1). ...* **[omitted 183 tokens]**

**Thought3**: **Alternatively,** *perhaps I can find a relationship between the sum S_n = sum_{k=1}^n f(k) and S_{n+1} ...* **[omitted 176 tokens]**

· 
·

**Thought 21**: **Alternatively,** *Alternatively, let me see if there's a closed formula.*

**Thought 22**: **Alternatively,** *note that:*
*Sum2 = sum_{k=1}^N [ (-1)^{k+1}(k +1) ....*

**Solution:**

Given the recurrence relation:

$$f(n + 1) = (-1)^{n + 1} n - 2f(n)$$

First, we express f(n) in terms of f(1). We find that:

$$f(n) = C(n) + (-2)^{n-1} f(1)$$

Next, we compute the

$$\text{sum ( } S\_N = sum\_\{k=1\}^N f(k) \text{ )}.$$

Substituting f(1) ,back into the expression for S_{1985} , we find:

$$S\_\{1985\} = \frac\{993\}\{3\} = 331$$

Final Answer: 331

**SmartSwitch(Our): ( 6012 tokens)**

**Thought1**: *Okay, so I need to solve this recurrence relation problem....*

*Starting with n = 1:\nf(2) = (-1)^{1+1} * 1 - 2f(1) = (-1)^2 * 1 - 2f(1). ... Hmm, this is getting a bit messy, but maybe there's a pattern here. ... Now let's look at the constant terms:\n\nFor n=1: ...Therefore, equate left and right:\n\nC(n+1) + (-2)^n f(1)...* **then maybe relate it using the recurrence?** ~~Alternatively,~~
**[potential score : 0.705 ]** *Dive Deep !!!*

**Thought2**: **Wait, let's dive deeper into this idea.** *Let me think about the recurrence relation and how it can relate the sum. ....*

*Let me consider the recurrence:f(n+1) + 2f(n) = (-1)^{n+1} *n. ... Therefore, solving for S_N: 3S_N = f(1) - f(N+1) + sum_{n=1}^N (-1)^{n+1}. ...Ah, right! The problem has the condition f(1) = f(1986). So, in my test case, ...* **make sure I didn't make any miscalculations here. Let me think again.** ~~Alternatively,~~
**[potential score : 0.723 ]** *Dive Deep !!!*

**Thought3**: **Wait, let's dive deeper into this idea.** *The key steps are .....*

· 
·

**Solution:**

Given the recurrence relation:

$$f(n + 1) = (-1)^{n + 1} * n - 2f(n)$$

The left-hand side can be simplified as:

$$3 * sum\_\{n=1\}^\{1985\} f(n) - f(1) + f(1986)$$

The right-hand side is the alternating sum:

$$sum\_\{n=1\}^\{1985\} (-1)^(n + 1) * n$$

Thus, we have:

$$3 * sum\_\{n=1\}^\{1985\} f(n) = 993$$

$$sum\_\{n=1\}^\{1985\} f(n) = 993 / 3 = 331$$

Final Answer: 331

Figure 8: Case study 3 (Math 500 recurrence). Left: vanilla QwQ-32B (correct, 11,244 tokens). Right: SmartSwitch-augmented QwQ-32B (correct, 6,012 tokens).

