# OpenReview forum: "SmartSwitch: Advancing LLM Reasoning by Overcoming Underthinking via Promoting Deeper Thought Exploration"
_ICLR.cc/2026/Conference — Submitted to ICLR 2026_

### Official Review · Reviewer_xqVs · 2025-10-27

**Soundness:** 3
**Presentation:** 3
**Contribution:** 3
**Rating:** 8
**Confidence:** 3

**Summary:**

This paper introduces SmartSwitch, a novel inference framework designed to address the "underthinking" problem in Large Language Models (LLMs) utilizing Long Chain-of-Thought (LongCoT) reasoning. The authors identify "underthinking" as a key limitation where models exhibit shallow reasoning, frequently and prematurely abandoning promising thoughts, which harms both performance and token efficiency. SmartSwitch operates as a plug-and-play, fine-tuning-free solution that continuously monitors the model's output. Its Perception module detects thought switches using linguistic cues (e.g., "Alternatively, ...") and uses an off-the-shelf Process Reward Model (PRM) to score the potential of the just-abandoned reasoning path. If a thought is deemed high-potential yet prematurely discarded, the Intervention module interrupts the generation, backtracks, and injects a "deepen prompt" to force the model to explore that promising path more thoroughly. Extensive experiments on challenging mathematical reasoning benchmarks, such as AIME24 and MATH-500, show that SmartSwitch significantly enhances the accuracy of various LLMs, for instance, boosting QwQ-32B by 10.0 points on AIME25. The framework effectively mitigates underthinking by reducing the number of thought switches, leading to deeper and more successful reasoning processes.

**Strengths:**

1. The paper addresses an important topic that influences LongCoT reasoning
2. The paper provides a thorough qualitative investigation of the phenomenon of the underthinking problem
3. The paper provides a good method that addresses the underthinking problem

**Weaknesses:**

1. The concrete algorithm for the proposed "adaptive paragraph" is not given in the paper, and the term is not mentioned in the main paper.
2. The baseline methods, Standard prompting and TIP (Thought Switching Penalty) (Wang et al., 2025), are neglected in the main comparison of Table 1
3. The work did not provide empirical evaluations on the hyperparameter settings.
4. The work did not provide an analysis of the operational process of the proposed method.

**Questions:**

1. How does the performance vary when the hyperparameters are adjusted? Can an optimal value be obtained? Does the performance vary over the repetition runs under the same hyperparameter setting?
2. During an inference, would the intervention model be called multiple times, or is there a higher probability of exiting the reasoning phase after an intervention?
3. Do thought processes that are deepened by intervention more often correspond to the final answer?
4. Would the performance change if the deepen prompt is adjusted to a different wording or instruction?

---

### Official Review · Reviewer_CSmZ · 2025-10-31

**Soundness:** 2
**Presentation:** 1
**Contribution:** 2
**Rating:** 2
**Confidence:** 4

**Summary:**

This paper identifies and addresses the "underthinking" problem in Long Chain-of-Thought (LongCoT) reasoning, where LLMs prematurely abandon promising reasoning paths, leading to shallow exploration and reduced performance. The authors propose SmartSwitch, a plug-and-play inference framework that monitors thought switches using linguistic cues, evaluates the potential of abandoned thoughts via an off-the-shelf Process Reward Model (PRM), and, when a high-potential thought is detected, intervenes by backtracking and injecting a "deepen prompt" to encourage further exploration. Evaluated on five mathematical reasoning benchmarks, SmartSwitch consistently improves accuracy across multiple LLMs, while also reducing token usage and inference time.

**Strengths:**

- Clearly defines and empirically validates the underthinking phenomenon across multiple models and difficulty levels.
- Proposes a simple, training-free, and model-agnostic framework that is easy to integrate.
- Demonstrates consistent and significant performance gains, especially for smaller models, and even improves efficiency (fewer tokens, faster inference).

**Weaknesses:**

- Reliance on explicit linguistic cues: The method detects thought switches only via predefined phrases (e.g., "Alternatively"), missing implicit or subtle shifts in reasoning. In contrast, concurrent work like SwiReasoning (arXiv:2510.05069) effectively handles implicit thought switching in latent space.
- Strong dependence on external PRM quality: Performance hinges on an off-the-shelf Process Reward Model. If the PRM is poorly calibrated, biased, or lacks sufficient context length, SmartSwitch may fail or even degrade reasoning.
- Risk of disrupting valid exploration: Forcing deeper exploration of a single path might suppress necessary perspective shifts, especially in problems requiring multiple complementary strategies.
- Hyperparameter sensitivity: Performance is highly sensitive to the manually tuned score threshold (τ = 0.7) and maximum intervention count (3); minor adjustments cause significant accuracy fluctuations.
- Limited comparison with concurrent work and narrow evaluation scope: The paper lacks comprehensive comparison with recent concurrent methods addressing underthinking or thought-switching, and all experiments are confined to mathematical reasoning.

**Questions:**

See the Weaknesses above.

---

### Official Review · Reviewer_FUPP · 2025-11-01

**Soundness:** 2
**Presentation:** 2
**Contribution:** 2
**Rating:** 4
**Confidence:** 4

**Summary:**

This paper studies underthinking in Long Chain-of-Thought (LongCoT) reasoning—i.e., models prematurely switching thoughts without sufficiently exploring promising lines—and proposes a test-time, training-free workflow called SmartSwitch. During generation, a Perception module detects thought switches (via surface cues) and scores the just-abandoned thought using a process reward model (PRM). If the score exceeds a threshold, an Intervention module backtracks and inserts a “deepen” prompt to encourage further exploration along that path. Experiments (mainly on competitive math benchmarks) show accuracy gains and some reductions in token usage/time compared to vanilla decoding.

**Strengths:**

- Clarity: The paper is generally well written; the two-stage Perception/Intervention pipeline and pseudocode are easy to follow, and figures convey the intended workflow.
- Problem framing: Surfacing underthinking—frequent, shallow thought switching—as a failure mode is an interesting observation that resonates with practitioner experience.
- Empirical signal: On several math benchmarks and across multiple model sizes, the proposed workflow reports non-trivial gains over vanilla inference, sometimes with improved efficiency.

**Weaknesses:**

1.	Evidence that mainstream LongCoT models “underthink” is not yet solid.

- Definition sensitivity. Underthinking is operationalized primarily by a length threshold on segmented “thoughts.” But different thought types (problem analysis, sanity check, verification, algebraic manipulation) naturally vary in token needs; short ≠ shallow. The current UF(L) metric therefore risks labeling by length, not by insufficient exploration. Please provide a typology of thought roles and token distributions per role to test whether specific roles are disproportionately short—and whether that correlates with failure.
- Segmentation confound. Thoughts are segmented by another strong LLM and linguistic cue rules; any segmentation error propagates into UF(L). You should quantify segmentation quality (inter-rater agreement vs. human annotation, or at least consistency across segmenters) and show robustness of UF(L) under alternate segmenters and thresholds.
- Model coverage. The claim that “underthinking is widespread” would be more convincing with deeper audits on state-of-the-art LongCoT-native models (e.g., DeepSeek-R1/R1-Distill, Qwen “thinking” variants) using the above stronger methodology rather than relying mainly on a token-length heuristic.

2.	Detector/decision rule is overly crude.

SmartSwitch flags switches using surface cues (e.g., “Alternatively…”) and then applies a fixed PRM threshold. This is a fairly brittle pipeline. To justify the simplicity, please measure recall/precision of (a) the switch detector and (b) the “promising-thought” classifier, using a high-capacity judge (e.g., GPT-5/o3-grade) or human annotation as reference. Report PR/ROC and error modes.

3.	Limited methodological novelty.

Injecting guidance prompts at test time to steer deeper exploration is a familiar idea in test-time steering/scaling. The main incremental aspect is the specific trigger (switch cue + PRM score). The paper would benefit from a crisper positioning against related TTS paradigms.

4.	Experimental scope is insufficient.

- Benchmarks: All results are math-only. Given the general framing (“underthinking”), evaluation should extend to code, STEM QA, and logic tasks to test generality.
- Baselines: As a test-time strategy, SmartSwitch should be compared against strong TTS baselines under matched token budgets: best-of-N / self-consistency (majority voting), diverse sampling + vote, tree/beam search, s1/other structured TTS, and decoding penalties tuned to the same cost. Also include iso-performance comparisons (achieve the same accuracy: who spends fewer tokens?).
- Budget accounting: While some tables suggest token/time reductions, a complete token-accounted budget (LLM tokens + PRM context + intervention restarts) per solved/unsolved item is needed to ensure the gains are not simply from spending more in aggregate.

5.	Causal attribution remains unclear.

It’s not shown that improvements come specifically from catching premature switches (the underthinking mechanism) versus generic “pause & encourage” effects. Controlled counterfactuals—e.g., trigger the deepen-prompt at random non-switch points with matched frequency; or intervene on obviously unpromising thoughts—would strengthen causal claims.

**Questions:**

1.	Validity of the UF(L) metric.

- Can you provide a role-aware audit of thoughts (analysis/derivation/checking/etc.) with token distributions and error correlations? Does underthinking persist after conditioning on role?
- How robust are the conclusions to different segmenters (LLM A vs. LLM B vs. human), thresholds, and cue lists?

2.	Detector evaluation.

- What are the recall/precision of switch detection and of “promising thought” classification against a strong adjudicator or human labels?
- How often does SmartSwitch mistakenly deepen genuinely bad thoughts, and what is the performance impact?

3.	Token-budget–matched TTS comparisons.

- Under equal token budgets, how does SmartSwitch compare to best-of-N/self-consistency, tree search, or structured TTS (e.g., breadth-limited expansions)?
- Conversely, at equal accuracy, which approach spends fewer total tokens (including PRM and restarts)? Please add these plots/tables.

4.	Causal tests of the mechanism.

- If you randomize intervention points or intervene on low-PRM thoughts with the same frequency, do you still see gains?
- Can you show that reducing measured underthinking (UF(L) or a better metric) mediates the accuracy gains?

5.	Beyond mathematics.

- Do the effects hold on coding, STEM QA, and logic benchmarks? If not, what failure modes emerge, and how should the detector/PRM be adapted?

---

### Official Review · Reviewer_QtZ4 · 2025-11-02

**Soundness:** 3
**Presentation:** 2
**Contribution:** 2
**Rating:** 2
**Confidence:** 5

**Summary:**

The paper targets underthinking in chain-of-thought (CoT) reasoning. It proposes a perception–intervention framework (SmartSwitch): a perception module detects potential switch cues (e.g., “Alternatively,” “On the other hand,” paragraph breaks, self-corrections) which typically appears in R1 or Qwen models and estimates whether the just-abandoned branch still has high potential.

If yes, an Intervention module backtracks and injects a brief deepening hint before continuing the generation. The method is training-free and aims to be model-agnostic. Experiments are reported mainly on math reasoning benchmarks with small to mid-size LMs, with ablations on cue lists, thresholds, and the number of allowed interventions.

**Strengths:**

1. Simple. training-free control that can be bolted onto existing decoders. Easy to prototype and tune.

2. Actionable diagnosis of underthinking, with concrete triggers and a backtracking-then-deepen intervention.

3. The method is motivated for SLMs where a little extra test-time guidance can yield meaningful gains under tight cost/latency budgets; this connects well to the hybrid SLM+LLM literature (e.g., selective teacher hints [A]).

Btw, it is good to include the related works around this area (including speculative reasoning fields).
[A] Guiding Reasoning in Small Language Models with LLM Assistance, Kim et al. (2025).

**Weaknesses:**

1. Perception dependence on discourse style / tokenizer. Cue lists like 'Alternatively' are common in R1/Qwen-style verbose reasoning but appear far less in GPT-OSS or proprietary models whose analysis channel is terse or hidden. This risks under-triggering on compact styles and over-triggering on verbose ones. Token-level signals (entropy spikes) are mentioned but not deeply validated. **Recommend more models outside of the Qwen styled".

2. Domain. Nearly all evidence is math-centric. It remains unclear whether the same cues and thresholds transfer to code (e.g. LiveCodeBench), science QA (e.g. GPQA), or tool-augmented tasks where “switching” has different surface forms.

3. Results would benefit from reporting variation (statistics) and a more explicit breakdown of overhead (PRM latency vs. decoding), to confirm that time reductions persist across settings.

4. Comparisons emphasize heuristic alternatives (e.g., always-intervene, different thresholds). Stronger test-time control baselines (e.g., verification-gated deepening, step-budgeted consistency, or learned switch detectors) would sharpen the claims. By the way, it's good to set the baseline with SMART [A], whose inspiration or motivation is quite similar but different approach.

[A] Guiding Reasoning in Small Language Models with LLM Assistance, Kim et al. (2025).

[Optional] 5. Generalization across languages. Non-English switching markers vary. The paper does not systematically study multilingual cue robustness.

**Questions:**

1. How does the perception module perform on models that expose fewer reasoning markers (e.g., compact CoT or hidden analysis channels)? Did you evaluate on GPT-OSS-style or proprietary models? Any failure cases or mitigations?

2. Did you use token log-prob/entropy spikes to detect implicit switches (no explicit cue)? How much does this improve precision/recall relative to text cues alone?

3. Multilingual and domain transfer. Do cue lexicons and thresholds transfer to KR/ZN, and to code/QA with tool use? If not, what changes?

4. Intervention budget. You cap interventions (e.g., ≤3). How do accuracy and latency trade off as the cap varies {1,2,4,∞}? Where is the knee of the curve?

5. Relation to hybrid SLM+LLM guidance. SMART-style selective teacher hints show small-model effectiveness with uncertainty-triggered interventions. Can your perception score gate such hints, and how would that compare to your PRM-gated deepening?

---

### Meta-Review · Area_Chair_ykdr · 2026-01-07

**Summary:**

This paper proposes SmartSwitch, a test-time framework that mitigates "underthinking" in LongCoT models by detecting premature thought switches via linguistic cues, scoring abandoned thoughts with a PRM, and prompting deeper exploration. The reviewers expressed concerns from the perspectives of problem characterization, methodological robustness, and experimental completeness, but the authors did not submit a rebuttal. Therefore, I suggest the authors continue to improve this paper to enhance its clarity, generality, and practical impact.

**Reviewer Concerns:**

The authors did not conduct a rebuttal, resulting in all the issues raised by the reviewers remaining unresolved.

**Reviewer Scores:**

Since no rebuttal was performed, the default score remains unchanged at 8 4 2 2.

---

### Decision · Program_Chairs · 2026-01-26

Reject